# Deciphering the Impact of Defecation Frequency on Gut Microbiome Composition and Diversity

**DOI:** 10.3390/ijms25094657

**Published:** 2024-04-25

**Authors:** Gwoncheol Park, Seongok Kim, WonJune Lee, Gyungcheon Kim, Hakdong Shin

**Affiliations:** 1Department of Food Science & Biotechnology, College of Life Science, Sejong University, Seoul 05006, Republic of Korea; gp21p@fsu.edu (G.P.); skim01@sejong.ac.kr (S.K.); leewonjune@sejong.ac.kr (W.L.); paulkc12@gmail.com (G.K.); 2Carbohydrate Bioproduct Research Center, Sejong University, Seoul 05006, Republic of Korea; 3Department of Health, Nutrition & Food Sciences, College of Education, Health & Human Sciences, Florida State University, Tallahassee, FL 32306, USA

**Keywords:** stool defecation, defecation frequency, gut microbiome, gut metabolome

## Abstract

This study explores the impact of defecation frequency on the gut microbiome structure by analyzing fecal samples from individuals categorized by defecation frequency: infrequent (1–3 times/week, *n* = 4), mid-frequent (4–6 times/week, *n* = 7), and frequent (daily, *n* = 9). Utilizing 16S *rRNA* gene-based sequencing and LC-MS/MS metabolome profiling, significant differences in microbial diversity and community structures among the groups were observed. The infrequent group showed higher microbial diversity, with community structures significantly varying with defecation frequency, a pattern consistent across all sampling time points. The *Ruminococcus* genus was predominant in the infrequent group, but decreased with more frequent defecation, while the *Bacteroides* genus was more common in the frequent group, decreasing as defecation frequency lessened. The infrequent group demonstrated enriched biosynthesis genes for aromatic amino acids and branched-chain amino acids (BCAAs), in contrast to the frequent group, which had a higher prevalence of genes for BCAA catabolism. Metabolome analysis revealed higher levels of metabolites derived from aromatic amino acids and BCAA metabolism in the infrequent group, and lower levels of BCAA-derived metabolites in the frequent group, consistent with their predicted metagenomic functions. These findings underscore the importance of considering stool consistency/frequency in understanding the factors influencing the gut microbiome.

## 1. Introduction

The gut microbiome, which comprises 10–100 trillion microorganisms, plays a crucial role in contributing to human health through individual or collective functions of these microbiota [1,2,3]. Human gut microbiota varies significantly among individuals, based on host properties, such as age, nationality, gender, or body mass index (BMI) [4]. These properties have become obstacles to studying the relationships between the human microbiome and health. To overcome this challenge, recent studies have introduced enterotyping, which stratifies multi-dimensional human gut microbial communities into three distinct enterotypes by the predominance of specific genera: *Bacteroides* (enterotype 1), *Prevotella* (enterotype 2), and *Ruminococcus* (enterotype 3) [4,5,6,7]. These enterotypes represent densely populated areas of community composition in a multi-dimensional space, aiming to simplify the complexity of the gut microbiota by reducing individual variation [4]. This approach has enabled the identification of the relationships and/or connections between the human microbiome and various diseases or diets, including the development of diagnostics for cancer and implications for weight loss, as well as the introduction of personalized nutrition in obesity management [8,9,10,11]. Previous reports have demonstrated that Koreans are clustered into two enterotypes, dominated by either *Bacteroides* or *Prevotella* [12,13].

A variety of factors associated with bowel movements, including the colonic transit time [14], and stool consistency [15], have been reported to affect the gut microbiome. Previous studies have shown that defecation frequency is considered as one of the factors that can affect the gut microbial community and the more frequent the defecation, the lower the diversity of the gut microbiota [16,17]. Given that previous studies have only considered single timepoint fecal samples and lacked a multi-omics approach, we aimed to comprehensively investigate the correlation between defecation frequency and the structure of the gut microbiome. We achieved this by incorporating bioinformatics data from a single participant, analyzed through longitudinal observations, along with metabolomic profiles.

In this study, we recruited 27 Korean volunteers, analyzing six longitudinal fecal samples from each subject to investigate the relationship between defecation frequency and gut microbial profiles. Consistent with previous reports, we observed that a higher defecation frequency was associated with a reduction in microbial diversity, and the predicted functional genes and the gut microbial structure varied depending on the defecation frequency.

## 2. Results

### 2.1. General Characteristics of the Subjects

Twenty adults, aged 22–31, with a body mass index (BMI) ranging from 18.42 to 27.47 kg/m^2^, were enrolled in this study. All subjects had not been prescribed antibiotics for one month prior to the experiment and were stratified into three groups based on their natural defecation frequency, as determined through a questionnaire: Group 1 (1–3 times/week, *n* = 4), Group 2 (4–6 times/week, *n* = 7), and Group 3 (everyday, *n* = 9). The defecation frequency remained consistent among all the subjects throughout the study period, which lasted up to 23 days. The sampling schedules are detailed in Appendix A. It should be noted that the stool collection interval varied among the subjects. However, our primary aim was to capture stable characteristics of the gut microbiome associated with defecation frequency over one month, rather than to track short-term fluctuations. Consequently, no statistical difference in the gut microbial structure was observed at any sampling timepoints within each defecation frequency group, as reflected by the intergroup distance (Appendix A). No significant differences were found in regard to age (one-way ANOVA test; *p*-value = 0.540) and BMI (*p*-value = 0.836), according to the defecation frequency groups. The general characteristics of all the participants are listed in Table 1.

### 2.2. Identification of Enterotypes among All Subjects

A total of 2,596,610 sequence reads were obtained from the fecal samples, with an average of over 20,000 sequence reads per sample. These reads were binned into 870 amplicon sequence variants (ASVs) (Table 2).

To stratify the population for a better understanding of the complexity of the gut microbiota by minimizing the impact of individual variability, the enterotype of the subjects was determined using baseline samples. The samples were clustered based on the relative abundance at the genus level using the Jensen–Shannon divergence (JSD) distance and the partitioning around medoids (PAM) clustering algorithm. The optimal number of clusters was estimated using the Calinski–Harabasz (CH) index (Figure 1A). The structures of the microbiota from all the samples were separated into two distinct clusters (Figure 1B; 27 subjects with a *Bacteroides*-dominant type and five subjects with a *Prevotella*-dominant type). To further corroborate the clustering across all the subjects based on their enterotypes, we also employed the Bray–Curtis distance metric. All the subjects were separated by their Bray–Curtis distance and marked with their respective enterotypes for the principal coordinates analysis (PCoA). Consistent with the JSD index shown in Figure 1B, a similar separation in the PCoA plot was observed (shown in PC1, 15.4% variance) using the Bray–Curtis index (Figure 1C). Considering that *Prevotella*-dominant and *Bacteroides*-dominant types uniquely feature a predominance of either *Prevotella* or *Bacteroides*, respectively, we expected a distinct *Bacteroides* and *Prevotella* ratio (PB ratio, *Prevotella*/*Prevotella* + *Bacteroides*) based on their enterotype classification. Accordingly, we calculated the PB ratio for each participant. While most subjects generally clustered into two distinct groups, several in the *Bacteroides*-dominant cluster (e.g., Sub. 083, 094, 107, and 109) showed ambiguous PB ratio values that set them apart from others in same cluster (Figure 1D). Considering that human gut microbiota varies greatly across individuals depending on host properties, which could skew data analysis, we decided to filter the subjects based on the following criteria: (1) those with distinct *Bacteroides* dominance (PB ratio < 0.02); (2) those whose microbiota dominance is maintained across all sampling timepoints. Accordingly, based on the criteria, only samples from a total of 20 subjects with the *Bacteroides*-dominant enterotype were used for further analysis and the consistency of the *Bacteroides*-dominant enterotype across all six sampling points for these subjects was confirmed (Appendix A).

### 2.3. Microbial Diversity and Structure Differences According to Defecation Frequency

To determine the relationship between microbial diversity and defecation frequency, the alpha diversity was measured using Faith’s PD and observed features indices. The microbial diversity of the ‘1–3 times a week’ group was significantly higher than in the other two groups for both metrics, while no difference was observed between the ‘4–6 times a week’ and ‘everyday’ groups. These results demonstrate that higher defecation frequency is associated with lower microbial diversity in the gut (Figure 2). In line with this trend, we also observed a negative relationship between microbial diversity and defecation frequency at each sampling timepoint, albeit with slight variability (Appendix A).

To evaluate the differences among the samples in regard to the bacterial communities, PCoA analysis based on the weighted UniFrac distance was performed, revealing that each group was well separated, and their community structures significantly differed depending on the defecation frequency (Figure 3A, PERMANOVA *p* < 0.001). The separation also remained consistent at each sampling timepoint (Appendix A). We then examined the pairwise distances between the groups, according to the defecation frequency: intragroup distances for the everyday samples were significantly greater than the other two groups (Figure 3B); the intergroup distance of the everyday group compared to the 1–3 times a week group was larger than that of 4–6 times a week; and the difference between the pairwise distance was shown to be statistically significant (Figure 3C). Consistent with the weighted UniFrac distance, a similar separation according to the defecation frequency was also observed using the unweighted UniFrac distance in the PCoA plot and a similar trend remained at each sampling timepoint (Appendix A).

### 2.4. Microbial Composition Difference and Relationship with Defecation Frequency

We observed differences in the microbial composition of the gut microbiota based on the defecation frequency. To further dissect which bacterial taxa are differentially prominent in each group, we analyzed the relative bacterial abundance and discriminant taxa using ANOVA-like differential expression 2 (ALDEx2) at the genus level. The *Ruminococcus* genus was significantly overrepresented in the ‘1–3 times a week’ group compared to the other two groups, and its abundance decreased as the defecation frequency increased. Although the *Ruminococcus* genera was identified as a significant component of the gut microbiota in our subjects, its presence alone was not sufficient to alter their established enterotypes (Appendix A). In contrast, the *Bacteroides* genus was highly prominent in the ‘everyday’ group compared to the other two groups, and its abundance decreased as the defecation frequency became less frequent (Figure 3D). Likewise, the same trend was also observed at each single timepoint (Appendix A). The ‘1–3 times a week’ group showed more overrepresented bacterial taxa when compared to the ‘4–6 times a week’ or ‘everyday’ groups, respectively (Figure 3E).

### 2.5. Predictive KEGG Functional Profiling

The Phylogenetic Investigation of Communities by Reconstruction of Unobserved States 2 (PICRUSt2) was used to predict the KEGG functional bacterial gene profiles from the 16S *rRNA* dataset in this study. To assess whether distinct bacterial genes are associated with defecation frequency, we performed a pairwise comparison between the infrequent (‘1–3 times a week’) and the frequent (‘4–7 times a week’) groups. Statistically significant pathways were selected if LDA > 2 and the FDR-adjusted *p*-value was <0.05. The infrequent group showed higher differential enrichments of bacterial genes related to the biosynthesis of fatty acids, vancomycin group antibiotics, peptidoglycan, branched-chain amino acids (BCAAs), lysine, aromatic amino acids, pantothenate and CoA, and terpenoid backbone. Additionally, genes associated with metabolism of propanoate, sulfur-containing amino acids, pyruvate, methane, histidine, glycerophospholipid, pyrimidine, nicotinate and nicotinamide, and retinol were, statistically, significantly more prevalent in the infrequent group when compared to the frequent group. In contrast, the frequent group exhibited a higher prevalence of the bacterial genes related to the metabolism of BCAAs, glyoxylate and dicarboxylate, glutathione, amino sugar and nucleotide sugar, vitamin B6, hexose, lipoic acid, biotin, and the biosynthesis of ubiquinone and other terpenoid quinone, folate, lipopolysaccharide, and polyketide sugar units. Notably, the interesting difference in the overlapping pathways between the two groups includes a higher enrichment of BCAA biosynthesis in the infrequent group and a higher prevalence of BCAA degradation in the frequent group (Figure 4A).

### 2.6. Differences in the Metabolites Related to Defecation Frequency

To investigate whether specific metabolites are associated with defecation frequency, we analyzed the metabolic profiles, using LC-MS/MS on the fecal samples. Six randomly selected samples from adults, aged 23–34, with a body mass index (BMI) ranging from 17 to 24 kg/m^2^, were included in the analysis. They were divided into two groups based on their defecation frequency: an infrequent group (‘1–3 times a week’, *n* = 3) and a frequent group (‘4–7 times a week’, *n* = 3). Consistent with the result that the abundance level of *Ruminococcus* was inversely related with that of *Bacteroides* as the defecation frequency increases (Figure 3D), the same tendency was observed in the participants for the metabolite analysis (Appendix A). In our dataset, 814 metabolites in negative mode and 891 metabolites in positive mode were identified using the Compound Discoverer software 3.3 SP2 (Figure 4B, gray squares; Appendix A), with 154 metabolites annotated with a KEGG ID from open databases, such as MZcloud, ChemSpider, KEGG, and the Human Fecal Metabolome database (Figure 4B, blue squares; Appendix A). Consistent with Figure 4A, the pathways for fructose and mannose/galactose metabolism and valine/leucine/isoleucine degradation were significantly more prevalent in the frequent group. In contrast, pathways for phenylalanine/tyrosine/tryptophan and valine/leucine/isoleucine biosynthesis were significantly dominant in the infrequent group (Appendix A). Among the metabolites associated with phenylalanine, tyrosine, and tryptophan metabolism, compounds such as pentose (a precursor for the TCA cycle), indole, tyrosine, p-cresol, phenylalanine, and phenylpropanoic acid showed significant enrichment in the infrequent group compared to the frequent group (Figure 4B,C; red squares). Similarly, among BCAAs, only leucine was statistically more prevalent in the frequent group, although the other two were not statistically significant (Figure 4B–D; red squares).

### 2.7. Random Forest Prediction

To test whether the defecation frequency of the samples can be predicted based on their microbial compositional features, a random forest classifier (RFC) model was employed using the ASVs table with 50 decision trees. The model generated an overall accuracy of 60%, surpassing the 45% baseline (random prediction without complex computations). Regarding the stool groups, the ‘everyday’ group showed the highest accuracy (78%), followed by the ‘4–6 times a week’ group (72%) (Figure 5A). The best performance was observed in the ‘4–6 times a week’ group (area under the curve, AUC: 0.83), followed by the ‘everyday’ group (AUC: 0.76) (Figure 5B). The ‘everyday’ group showed the best separation in the random forest class probability histogram (Figure 5C).

## 3. Discussion

This study involved collecting longitudinal fecal samples from each individual over three weeks (six samples), instead of collecting them at a single timepoint. This approach helps control the variations in gut microbiota that could occur within an individual. Our results show clear and lasting differences in the gut microbial profiles according to the defecation frequency. We have demonstrated that less frequent defecation is associated with a richer population of microbes in the gut (Figure 2). In addition, the microbial structure is distinct, depending on the defecation frequency, based on weighted/unweighted UniFrac distances (Figure 3A and Appendix A), consistent with previous reports [16,17,18]. These results strongly indicate differences in either the microbial composition or abundance depending on the defecation frequency. It was interesting to observe the gradual movement of the samples from the infrequent to the frequent defecation group towards the PC1 axis, suggesting the possibility that the ‘4–6 times a week’ group has intermediate microbial structure characteristics between the ‘1–3 times a week’ and ‘everyday’ groups (Figure 3A). We found two different taxa that are differentially present depending on the defecation frequency: the *Ruminococcus* and *Bacteroides* genus (Figure 3D,E). Considering that intragroup distances in the ‘1–3 times a week’ and ‘4–6 times a week’ groups are not significantly variable, these specific taxa could potentially be considered as biomarkers for stratifying groups according to defecation frequency.

An overrepresentation in the methane metabolism pathway was observed in the ‘1–3 times a week’ group compared to the ‘4–6 times a week’ group (with no observation in the ‘everyday’ group) (Figure 4). This is in line with the fact that increased methane production was observed in people with constipation [18,19]. Previous studies have demonstrated the limited bioavailability of polysaccharides, as well as extensive protein catabolism as the retention time increases in in vitro models of the gut [20,21]. In addition to previous findings, recent studies have revealed the relationship between the colonic transit time (CTT), a proxy for defecation frequency, and bacterial metabolism, where a longer CTT is associated with higher colonic protein catabolism, as reflected by the presence of urinary metabolites like *p*-cresol sulfate and *p*-cresol glucuronide. These metabolites are products of tyrosine metabolism, as identified by urinary metabolic phenotyping in the gut [14,18,22]. Furthermore, a longer CTT is positively associated with the higher enrichment of *Ruminococcaceae* and *Ruminococcus* [14]. In this regard, it is noteworthy that the predicted functional profile related to tyrosine metabolism was overrepresented in the ‘1–3 times a week’ group compared to the ‘4–6 times a week’ group (Figure 4). In agreement with previous findings, our data also showed that the *Ruminococcus* genus and *Ruminococcaceae* family were significantly overrepresented in the ‘1–3 times a week’ group, and the functional genes involved in tyrosine metabolism were highly enriched in the ‘1–3 times a week’ group compared to the ‘4–6 times a week’ group (with no observation in the ‘everyday’ group). To address the limitation of genomic analysis based solely on 16S *rRNA* data, metabolomic analysis was further employed to support the functional gene predictions. Metabolite profiling indicated a higher enrichment of tyrosine metabolism in the infrequent group, evidenced by the increased level of metabolites involved in phenylalanine/tyrosine/tryptophan metabolism. In contrast, the frequent group showed a decreased level of branched-chain amino acids (valine, leucine, and isoleucine), whereas the infrequent group exhibited high levels of these amino acids (Figure 4B–D). These metabolic profiles strongly corroborate the functional KEGG pathway predicted using 16S *rRNA* data (Figure 4B–D). By integrating metabolite profiling with taxa analysis, we observed an overrepresentation of indole and *p*-cresol in the infrequent group, where the abundance of *Bacteroides* was low and that of *Ruminococcus* was high, compared to the frequent group (Figure 4 and Appendix A). These findings are in line with previous findings, demonstrating a negative correlation between the abundance of *Bacteroides* and the levels of *p*-cresol and indole, and a positive correlation with *Ruminococcus* (Figure 3 and Figure 4; [14,23,24]). Considering that *p*-cresol and indole, known as uremic toxins, can potentially cause chronic kidney diseases and cardiovascular diseases, and as such should be detoxified in the body [25,26], frequent stool defecation could be preferable for health. Previous studies have demonstrated that circulating BCAA levels are positively correlated with the development of obesity, and *Bacteroides* spp. are shown to promote BCAA catabolism in brown fat, thus preventing obesity [27,28]. Our findings, which indicate a higher abundance of *Bacteroides* and decreased levels of BCAA in the frequent group compared to the infrequent group, are in strong alignment with these studies. 

A strength of this study is the use of longitudinal sample collection from each individual to validate the connection between defecation frequency and gut microbiota, unlike previous studies that considered a single sample at one timepoint. We also employed in silico analysis, using a machine learning model (random forest) trained on the composition of the gut microbiota from individuals to build a prediction model. However, the limitations of this study include an insufficient number of subjects for analyzing *Prevotella*-dominant enterotypes, as well as metabolite analysis. Since each enterotype establishes distinct ecological niches, prolonged defecation frequency may induce markedly varied changes in microbial communities and their metabolic products. Futures studies with *Prevotella*-dominant enterotype subjects will help establish a correlation between different microbial compositions/metabolites and defecation frequency. To conduct a more comprehensive analysis of the relationship between defecation frequency and the microbiome profile, future studies should aim to assess defecation frequency using numerical values rather than categorical ones. This approach would facilitate correlational analysis between defecation frequency and microbial features, including microbial diversity and composition. To deepen our understanding of the comprehensive relationship between defecation frequency and the gut microbiome, further studies employing a comprehensive multi-omics approach with a lager participant cohort are necessary.

## 4. Materials and Methods

### 4.1. Sample Collection

Participants were instructed to self-collect fecal samples, twice weekly, over three weeks (23 days at most), at their convenience. Despite this schedule, delays in the collection of these samples, ranging from 1 to 4 days, were recorded for six participants. Sterile cotton-tipped swabs were used for the collection of fresh fecal samples, which were then immediately frozen at −80 °C until sequencing. For metabolite analysis, the fecal samples were self-collected by the participants using sterile swabs and subjected to immediate freezing at −80 °C until analysis.

### 4.2. The 16S rRNA Amplicon Sequencing

Total genomic DNA was extracted from the stool samples using the DNeasy PowerSoil HTP 96 kit (Qiagen, Hilden, Germany), according to the manufacturer’s instructions, and stored at −80 °C for subsequent experiments. The V4 hypervariable region of the 16S rRNA gene was amplified using 515F/806R primers [29]. Amplicons were purified using the NucleoSpin PCR clean-up kit (Macherey-Nagel, Düren, Germany) and subjected to sequencing using the Illumina MiSeq platform (2 × 300 cycles, paired end). Sample preparations and sequencing steps were performed according to the Earth Microbiome Project (www.earthmicrobiome.org) accessed on 16 February 2023.

### 4.3. Data Analysis

Analysis of the 16S rRNA sequences was performed using the Quantitative Insights Into Microbial Ecology (QIIME) software package 2-2020.6 [30]. Raw sequencing reads were demultiplexed and filtered on quality using the q2-demux plugin, followed by quality trimming and denoising with DADA2 [31]. All the produced amplicon sequence variants (ASVs) were aligned using MAFFT [32] and used to generate a rooted phylogenetic tree with FastTree 2 [33] for phylogenetic diversity analysis. The q2-feature-classifier plugin was used to trim the ASVs that were not from bacteria using the SILVA 132 database [34,35]. Each ASV was classified taxonomically using a naïve Bayes taxonomy classifier, implemented using the q2-feature-classifier plugin [36]. Faith’s phylogenetic diversity [37] and observed features were calculated to measure the alpha diversity, and the unweighted [38]/weighted [39] UniFrac distance were used for the beta diversity. Multiple statistical analyses were performed: (1) a non-parametric Kruskal–Wallis test [40] was used to determine significant differences in the microbial diversity; (2) permutational multivariate analysis of variance (PERMANOVA) was used to evaluate the difference in the community structure (999 random permutations) [41]; (3) a non-parametric *t*-test with 128 Monte Carlo instances was used to run ALDEx2, a tool that detects differentially abundant taxa between samples [42].

### 4.4. Enterotype Analysis

Baseline samples were clustered based on genus-level collapsed relative abundance using the Jensen-Shannon divergence (JSD) and the partitioning around medoids (PAM) clustering algorithm. The Calinski–Harabasz (CH) index [43] was assessed to determine the optimal number of clusters. The statistical significance of the optimal clustering was evaluated using the silhouette validation technique [44]. Enterotype analysis was performed in the R environment [4], and more detailed information is available at https://enterotype.embl.de/enterotypes.html accessed on 1 December 2023.

### 4.5. Predictive KEGG Function Profiling

The functional prediction of metagenomics profiles from the 16S rRNA gene sequence was computed using the Phylogenetic Investigation of Communities by Reconstruction of Unobserved States 2 (PICRUSt2) [45]. The ASVs table was used as input into the q2-picrust2 plugin of QIIME2. For the inference from the PICRUSt2 results, the Kyoto Encyclopedia of Genes and Genomes (KEGG) ortholog database [46] was used. Significant differences in the predicted functional metagenomic profiles were identified using linear discriminant analysis effect size (LEfSe) analysis (LDA score) [47].

### 4.6. Sample Preparation for Global Metabolomics

The fecal sample (500 μL) was extracted by mixing an equal volume of 10% of acetonitrile, followed by vigorous shaking for 1 min, then incubated at −20 °C for 2 h. The resulting mixture was centrifuged at 4 °C and 13,000× *g* for 15 min, followed by homogenization using a Tissue Lyser (Qiagen) at 30 Hz for 10 min. The supernatant was subjected to LC-MS/MS to profile the metabolites, with quality control (QC) samples used in parallel.

### 4.7. U-HPLC-MS/MS Conditions

The U-HPLC-MS/MS analyses were performed using a U-HPLC system (Vanquish^TM^ U-HPLC, Thermo Fisher Scientific, Waltham, MA, USA), coupled to a quadrupole mass spectrometer (Thermo Scientific Orbitrap Exploris^TM^ 120 high-resolution/accurate mass spectrometer, interfaced with a heated electrospray ionization (H-ESI) source), at the Biopolymer Research Center for Advanced Materials (BRCAM, Sejong University, Seoul, Republic of Korea). For the chromatographic separation, the prepared samples (10 μL) were injected into a C18 column (Waters, ACQUITY UPLC BEH C18, 2.1 × 100 mm, 1.7 μm, Waters Corp., Milford, MA, USA), with the column temperature maintained at 45 °C throughout the acquisition period. The mobile phase was eluted by a gradient of water (A) and acetonitrile (B), containing 0.1% acetic acid, with a gradient dilution profile of 95% A (0–2 min), 95–5% A (2–15 min), 5% A (15–17 min), 5–95% A (17–18 min), and 95% A (18–20 min), at a flow rate of 0.3 mL/min. The mass spectrum conditions included a heated capillary of 320 °C, a vaporizer temperature of 350 °C, a spray voltage of 3.5 KV in positive mode, and 2.5 KV in negative mode; the sheath gas, aux gas, and sweep gas were set at 50, 25, and 1 psi, respectively. The HCD collision energy was set at 15, 30, and 60%, and the scan range covered 55–700 *m*/*z* in full scan, with a resolution of 120,000 for MS1 and 15,000 for MS2.

### 4.8. Global Metabolomic Analysis

The raw MS data were processed using Xcalibur version 4.6 and Compound Discoverer 3.3 (Thermo Fisher Scientific, Waltham, MA, USA). Normalization of unknown compound intensities using the QC sample was performed. Enrichment analysis to identify the potential KEGG pathway was performed using MetaboAnalyst (https://www.microbiomeanalyst.ca/) accessed on 29 December 2023. Statistical significance was assessed using Student’s *t*-test, with *p*-values < 0.05 considered statistically significant.

### 4.9. Random Forest

For the random forest classifier prediction, supervised classification of the ASVs table generated from the baseline samples of each individual was performed using the q2-sample-classifier [48] plugin, via the nested stratified 4-fold cross-validation [49] classifier grown with 50 trees.

## Figures and Tables

**Figure 1 ijms-25-04657-f001:**
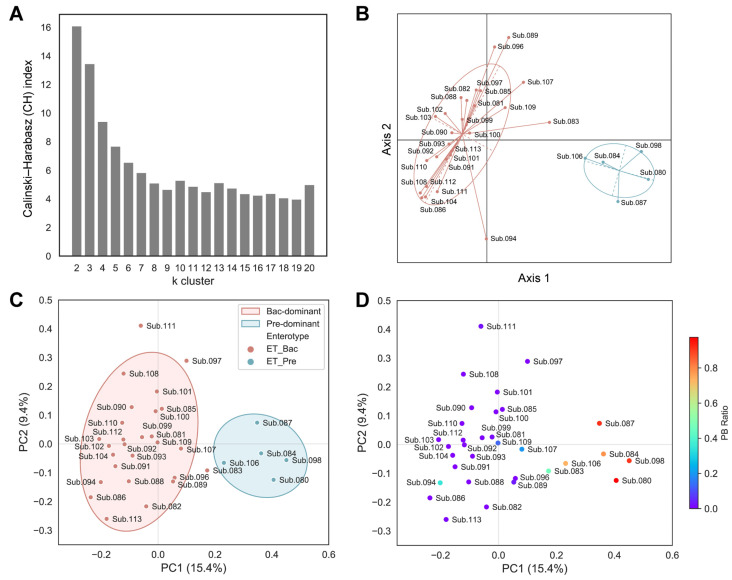
Gut enterotype classification of subjects. (**A**) Evaluated Calinski–Harabasz (CH) index for every number of k clusters to determine the optimal number of clusters. (**B**) Principal coordinates analysis (PCoA) plot of enterotyping data based on relative genus abundance using the Jensen–Shannon divergence (JSD) distance matrix and the partitioning around medoids (PAM) clustering algorithm. (**C**,**D**) PCoA plots based on the Bray–Curtis distance metric. (**C**) Samples are colored according to the enterotyping results. Covariance ellipses were projected for each cluster and the bounds of the cluster were marked by two standard deviations (2σ) in each direction from the mean of the cluster. (**D**) Samples are colored according to the PB ratio (*Prevotella*/*Prevotella* + *Bacteroides*). Coloring toward red or purple indicates *Prevotella*-dominant or *Bacteroides*-dominant subjects, respectively.

**Figure 2 ijms-25-04657-f002:**
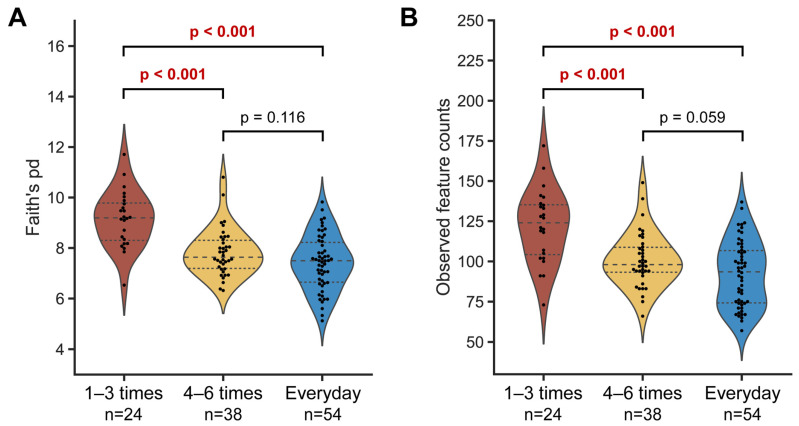
Alpha diversity of gut microbiota according to defecation frequency. Faith’s phylogenetic distance (**A**) and observed features metrics (**B**) were used to plot the graphs. Each single dot represents an individual. The dashed line represents the median and the dotted lines denote interquartile ranges (IQRs). Statistical analysis was conducted using the pairwise Kruskal–Wallis test.

**Figure 3 ijms-25-04657-f003:**
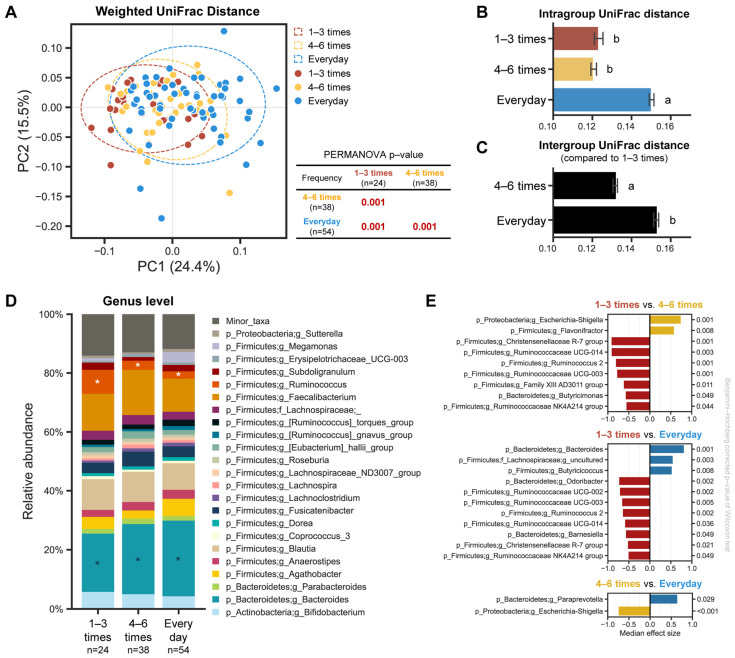
Microbial structure and composition of each group, depending on the defecation frequency. (**A**) PCoA plot based on weighted UniFrac distance. PERMANOVA was used to test the statistical differences among the groups. Intragroup (**B**) and intergroup (**C**) distances, using weighted UniFrac distance, across the three different groups. Data shown and error bars are mean ± SEM (non-parametric *t*-test, *p* < 0.001). Different alphabets indicate significant differences between the groups (*p* < 0.001). (**D**) Relative abundance plot of bacterial taxa at the genus level across all three groups. Asterisks (white/blue) denote differentially abundant taxa, according to the defecation frequency. (**E**) Bar chart showing differentially abundant taxa between the groups using ALDEx2 (median effect size > 0.5 and Benjamini–Hochberg corrected *p*-value of Wilcoxon test < 0.05).

**Figure 4 ijms-25-04657-f004:**
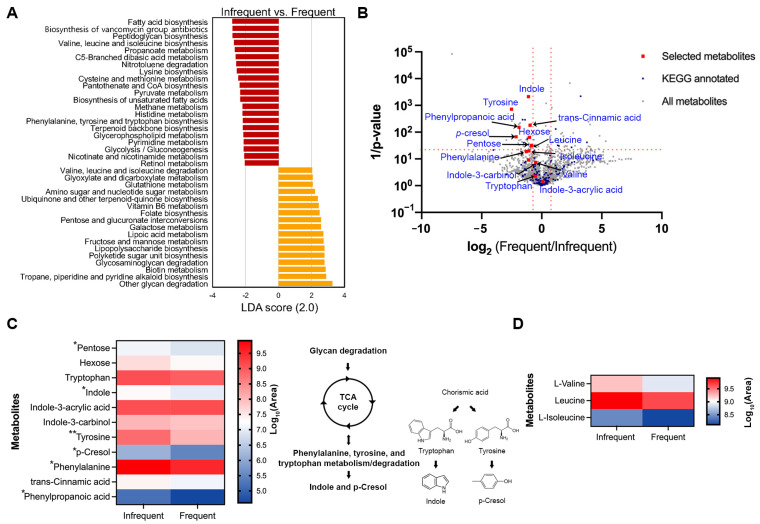
Predictive KEGG functional pathways that are differentially present in pairwise comparisons. (**A**) Bar chart represents distinctly abundant KEGG pathways that were selected based on the following criteria: LDA > 2 and adjusted *p*-value < 0.05. (**B**) Volcano plots of differential metabolites between the infrequent and the frequent group. All metabolites and those with KEGG annotations are indicated by gray and blue squares, respectively. The metabolites that overlap with predictive functional KEGG pathways generated using 16S *rRNA* data are shown in red squares. Metabolites were identified as candidates based on a log_2_(Frequent/Infrequent) ratio > 1 and statistical significance (*p* < 0.05) based on unpaired t test after two–stage step–up correction method (indicated by red dotted lines). (**C**,**D**) Heatmaps of differential metabolites in phenylalanine/tyrosine/tryptophan biosynthesis and degradation (**C**), and in branched-chain amino acid metabolism (**D**). Metabolites with asterisks are statistically significant based on unpaired t test after two–stage step–up correction; * *p* < 0.05; ** *p* < 0.01.

**Figure 5 ijms-25-04657-f005:**
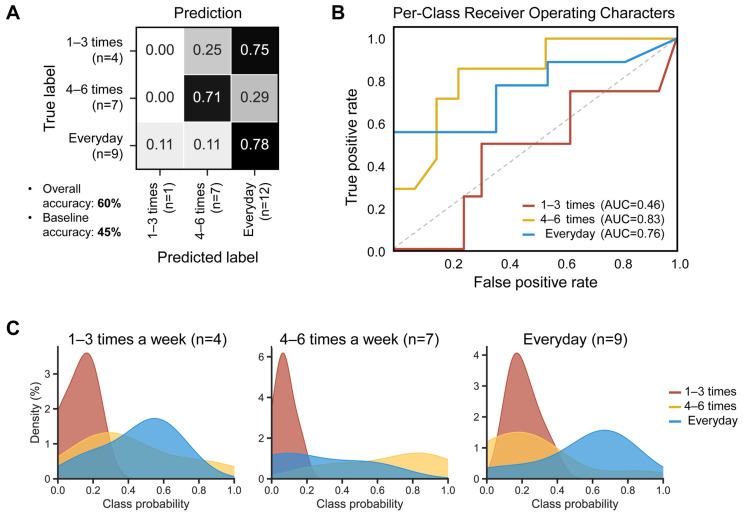
Diagnostic accuracy of gut microbial structure using the baseline fecal samples, according to defecation frequency. The higher the accuracy, the darker the shade. (**A**) The confusion matrix of the random forest classifier. (**B**) Receiver operating characteristic (ROC) curves, evaluating ability to predict defecation frequency using random forest classification. Classification accuracy was assessed by the area under the curve (AUC). The dashed line represents the points of no discrimination on a ROC curve. (**C**) Random forest class probability histograms for ‘1–3 times a week’ (red), ‘4–6 times a week’ (yellow), and ‘everyday’ (blue) according to the defecation frequency.

**Table 1 ijms-25-04657-t001:** General characteristics of subjects included in this study (age, sex, BMI).

		1–3 Times a Week	4–6 Times a Week	Everyday	*p* Value
Total no. of subjects		*n* = 4	*n* = 14	*n* = 14	-
No. of enrolled subjects		*n* = 4	*n* = 7	*n* = 9	-
Sex	Male, n (%)	2 (50.0)	1 (14.3)	5 (55.6)	-
	Female, n (%)	2 (50.0)	6 (85.7)	4 (44.4)	-
Age, average ± SD		24.0 ± 1.2	24.7 ± 1.7	23.9 ± 1.2	0.540
BMI, average ± SD		22.1 ± 1.5	21.3 ± 3.0	21.4 ± 1.1	0.836

**Table 2 ijms-25-04657-t002:** Reads and ASV counts for each group.

Defecation Frequency	1–3 Times	4–6 Times	Everyday	Total
No. of samples	24	38	54	116
No. of sequences	507,975	865,445	1,223,190	2,596,610
Average sequence ± SD	21,166 ± 4922	22,775 ± 5060	22,652 ± 3901	22,385 ± 4569
No. of features (ASVs)	459	486	532	870
Average features (ASVs) ± SD	124 ± 23	102 ± 17	94 ± 21	103 ± 23

## Data Availability

All amplicon sequence data and metadata have been made public through the EMP data portal (Qiita, https://qiita.ucsd.edu; Study ID: 15542) accessed on 23 April 2024.

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
