# Peer review of "Deciphering the Impact of Defecation Frequency on Gut Microbiome Composition and Diversity"

_ijms, 2024, doi:10.3390/ijms25094657_

Round 1
Reviewer 1 Report
Comments and Suggestions for Authors
Comments on the Quality of English LanguageSufficient, only minor corrections needed
Reviewer 2 Report
Comments and Suggestions for Authors
I think that this is a nice study with a lot of bioinformatics and not very much of biological analysis. It is interesting to read about the enterotypes of the group under study but there is no association between the enterotypes and stool frequency. Or may be it does exists but was not analyzed. Than what was the reason to analyze enterotypes which did not appear not in the introduction or results. It is interesting that Ruminococcus enterotype did not appear in the study but later the authors mentioned that infrequent group was associated with this enterotype. Regarding the experimental setup I would not suggest to call the people in the group with infrequent a healthy volunteers, because it is hard to call them really healthy according to all gastrointestinal criteria. Metabolic part of the study looks nice and convincing however the number of participants might be low.
